

# A new phylogenetic data standard for computable clade definitions: the Phyloreference Exchange Format (Phyx)

Gaurav Vaidya[1,2], Nico Cellinese[2,3] and Hilmar Lapp[4]

[1] Renaissance Computing Institute (RENCI), University of North Carolina at Chapel Hill, Chapel Hill, NC, United States of America
[2] Florida Museum of Natural History, University of Florida, Gainesville, FL, United States of America
[3] Informatics Institute, University of Florida, Gainesville, FL, United States of America
[4] Department of Biostatistics and Bioinformatics, Duke University, Durham, NC, United States of America

## ABSTRACT

To be computationally reproducible and efficient, integration of disparate data depends on shared entities whose matching meaning (semantics) can be computationally assessed. For biodiversity data one of the most prevalent shared entities for linking data records is the associated taxon concept. Unlike Linnaean taxon names, the traditional way in which taxon concepts are provided, phylogenetic definitions are native to phylogenetic trees and offer well-defined semantics that can be transformed into formal, computationally evaluable logic expressions. These attributes make them highly suitable for phylogeny-driven comparative biology by allowing computationally verifiable and reproducible integration of taxon-linked data against Tree of Life-scale phylogenies. To achieve this, the first step is transforming phylogenetic definitions from the natural language text in which they are published to a structured interoperable data format that maintains strong ties to semantics and lends itself well to sharing, reuse, and long-term archival. To this end, we developed the Phyloreference Exchange Format (Phyx), a JSON-LD-based text format encompassing rich metadata for all elements of a phylogenetic definition, and we created a supporting software library, phyx.js, to streamline computational management of such files. Together they form a foundation layer for digitizing and computing with phylogenetic definitions of clades.

## INTRODUCTION

As many other scientific disciplines, biology has become inundated with data. This includes biodiversity science and fields related to it. For example, species occurrence databases and repositories of digitized museum specimens house hundreds of millions of records (*Ariño, 2010*; *Page et al., 2015*). Integrating such data with genomic, phenomic and other biological data can be very powerful for discovering and understanding large-scale patterns (*Beaman & Cellinese, 2012*; *Ellwood et al., 2020*), but doing so requires the use of computer programs, *i.e.,* machines, that connect records by shared entities (*Parr et al., 2012*; *Heberling et al., 2021*). The most common shared entity among biodiversity data is the taxon, or more

Corresponding author
Hilmar Lapp, hilmar.lapp@duke.edu

specifically the taxonomic concept (*Berendsohn, 1995*) to which the data or specimens were identified by those who collected them.

Such identification is almost always recorded as a Linnaean name, which serves as a proxy for the taxonomic concept it points to. However, Linnaean names suffer from a number of shortfalls that are inherent in their nature and make them highly problematic for computation-driven data science. Chiefly, a Linnaean name is a mere text string decoupled from its original concept, which over time is subject to changing interpretations, making its meaning (semantics) often difficult or impossible to reconcile between different data sets. Even where a Linnaean name can be traced unequivocally to a published taxon concept, its semantics are not accessible to computation, and thus full computational reproducibility (*i.e.,* without the need to invoke human expert judgement) is often impossible to achieve (*Cellinese, Conix & Lapp, 2022*).

In contrast, the advent of tree-thinking (*Zimmermann, 1934*; *Hennig, 1950*; *Hennig, 1966*) gave rise to the establishment of phylogenetic systematics, and subsequently the development of phylogenetic taxonomy (*De Queiroz & Gauthier, 1990*; *De Queiroz & Gauthier, 1992*; *Queiroz, 1992*). A phylogeny is a hierarchical system of nested clades, where a clade is a group that includes an ancestor and all of its descendants. Within this framework, clades, not Linnaean groups, represent natural taxa. Phylogenetic definitions are an approach to unambiguously define taxa by shared ancestry rather than by trait similarity. Defining taxa in this way has been increasingly adopted in the wild, and a nomenclatural code (called the "PhyloCode") governing rules for definitions and naming has been established as well (*De Queiroz & Cantino, 2020*). Phylogenetic definitions come in the following three basic types (Fig. 1): (1) minimum clade definitions, which designate the smallest clade that originates with the most recent common ancestor of two or more internal specifiers; (2) maximum clade definitions, which designate the largest clade that includes one or more internal specifiers but excludes one or more external specifiers; and (3) apomorphy-based definitions, which designate the clade that arises from the first appearance of a specified trait that is synapomorphic with an internal specifier. Specifiers are reference points in the phylogeny that serve as anchors for the clade definition. They can be references to a taxon concept, such as a species, specimen, or a molecular sequence, or they can also be an apomorphy.

In recent years, very large phylogenetic trees, including the synthetic Open Tree of Life (*Hinchliff et al., 2015*), have become available as the basis for aggregating biodiversity data for comparative biology questions. Integrating Linnaean taxon concepts, which are defined by trait similarity, not ancestry, with a phylogeny requires repurposing them for nodes on a tree (*Cellinese, Baum & Mishler, 2012*), a fraught process for which verifying the match in semantics will be difficult or impossible. In contrast, phylogenetic definitions are native to tree-thinking (*Cellinese, Conix & Lapp, 2022*). Although usually published as unstructured natural language text, they lend themselves to transformation to structured data and to formal logic expressions that allow machines to understand and compute with their semantics. As a consequence, they can enable automated computational verifiability (do two taxon concepts mean the same thing?) and thus reproducibility in large-scale biodiversity data integration.

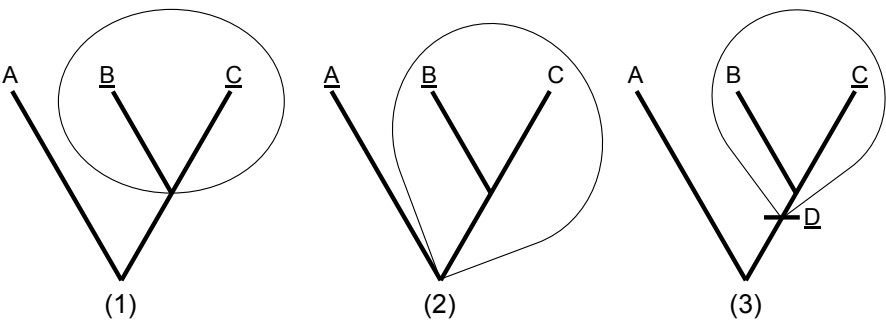

**Figure 1** **The three basic types of phylogenetic definitions.** (1) Minimum clade definitions, which designate the smallest clade that includes at least two internal specifiers (in this case, 'B' and 'C'); (2) Maximum clade definitions, which designate the largest clade that includes one or more internal specifiers ('B') but excludes one or more external specifiers ('A'), and (3) apomorphy-based definitions, which designate the clade that arises from the first appearance of a specified trait ('D') that is synapomorphic with an internal specifier ('C'). Redrawn from *De Queiroz & Gauthier (1990)*.

We have created a standard, called Phyloreferencing, for representing clade definitions with fully machine-processable semantics, using the Web Ontology Language (OWL) (*W3C OWL Working Group, 2012*). We refer to such representations as phyloreferences, in analogy to georeferences as fully computable geographic locations. We have also created supporting tools, methods, and a structured data exchange format for digitizing phylogenetic definitions from their original natural language text form to a phyloreference. In this paper we describe the structured exchange format, which we call Phyloreference Exchange Format (Phyx), its supporting software library (phyx.js), and tools and methods we use for automatic testing of whether the encoded semantics of a phyloreference indeed match (and only match) the expected clade on a phylogenetic tree. In Methods we explain the rationale for digitizing clade definitions first to the structured exchange format rather than directly to OWL, and give the requirements for the design of the format to properly support the digitization process as well as automatic testing of the resulting phyloreference records. In Results we present the Phyx format, the principal algorithm for converting Phyx records to an OWL representation of phyloreferences, the phyx.js supporting software library, and how we conduct automatic correctness testing of the OWL conversion result. Taken together, the results presented here provide a reusable foundation layer for digitizing clade definitions in an intermediate format that is easy to work with, suitable for technology-agnostic permanent archival, and which can be converted into computable, testable, semantically-rich OWL ontologies. Most of the resources we present here, including in particular the Phyx data standard and its supporting software library, are primarily aimed at developers building tools, although some tools directly suitable for end-users are included, such as for resolving phyloreferences against the Open Tree of Life.

## METHODS

### The need for a structured exchange format

OWL ontologies are ideally suited for stating the semantics of the clade definition as logical expressions, as well as for associating provenance and other metadata of the clade definition, such as authorship information, citations to published forms of the clade definition, and information about the reference phylogeny. OWL ontologies are also ideal for making the logical clade definition testable: with phyloreferences and their reference phylogenies in the same OWL ontology, an OWL reasoner could compare the resolution of a phyloreference on a reference phylogeny with the expected resolution as previously recorded by a curator, ensuring that the logical definition of a phyloreference continues to correctly represent the semantics of the corresponding clade definition.

In the typical life cycle, phyloreferences are not instantiated from scratch as logically defined OWL ontology classes, but are the product of digitizing a clade definition published in natural language text. As such, they form a complex data structure, and a format most suitable and versatile for their exchange, databasing, and long-term digital archival should meet the following desiderata:

- Text-based and editable with a non-specialized basic text editor
- Support for hierarchical, nested, key-value structures, where values can be of different data types
- Ability to include phylogenies in standard and widely supported formats, in particular Newick (https://evolution.genetics.washington.edu/phylip/newicktree.html)
- Validatable for correct syntax, so syntactical errors can be detected early and automatically
- Reading, writing, and manipulation well supported in a wide array of programming languages, in particular those used for programming the web
- Standard toolchain and specification for converting to a semantic web format, specifically the Resource Description Format (RDF) (*Brickley & Guha, 2014*) and OWL.

Obviously, the last of these would be a given with OWL as the digitization product and exchange format, but the serialization formats of OWL (see *W3C OWL Working Group, 2012*) are all a poor match for the other criteria. We therefore designed an intermediate format for digitizing clade definitions that would have all the information needed to generate a full OWL ontology, including the logical expressions that capture a clade definition's precise semantics. Given the desiderata enumerated above, we based this format on JSON, more specifically JSON-LD (*Sporny et al., 2014*), and named it the *Phyloreference Exchange* (or *Phyx*) Format.

JSON is the native data exchange format for JavaScript. It is also a widely used and well-supported ISO standard data exchange format (https://tools.ietf.org/html/rfc8259), and allows leveraging existing software support libraries and tools in many programming languages, including using JSON Schema (*Jackson, 2016*) to validate a particular JSON file for compliance with a schema. JSON-LD, which parses as JSON but has some conventions layered on top that allows assigning semantics to property keys and their values, is one of
the serialization formats for OWL ontologies (see *Sporny et al., 2020*), thus providing for a standard conversion path. To convert a Phyx file to an OWL ontology in JSON-LD format requires adding only a few additional components:

- Metadata on the ontology itself, such as which other ontologies to include.
- The conversion of phyloreferences from key-value pairs into OWL logical expressions.
- The conversion of phylogenies from the Newick format into an ontological representation that can be understood by an OWL2 reasoner.

To simplify many routine programming tasks when working with Phyx files, and to better control that the Phyx JSON-LD is in proper form for converting to an OWL ontology (such as for all entities generating and assigning unique IRI identifiers if needed), we created a supporting software library for the format, phyx.js (see Results).

## Minimum requirements for digitizing phyloreferences

To ensure that a Phyx record can be converted to a fully defined OWL class expression for the phyloreference, a certain minimum amount of information must be present. We determined that this minimum information is constituted by the clade type and the list of included and excluded taxonomic units (*i.e.,* the "specifiers"), and identified four kinds of specifiers required to be supported:

1. A taxon concept, represented either by an IRI (https://tools.ietf.org/html/rfc3987) or by a combination of a scientific name and an optional publication to indicate which circumscription is indicated.
2. A specimen, represented by a specimen identifier or an IRI.
3. An apomorphy, represented by a textual description or an IRI.
4. Any other definition for a taxon, such as another clade definition, represented by an IRI.

Each phyloreference should also allow additional provenance metadata to be recorded, such as information about the authorship and publication status of the Phyx file, the phyloreferences, or the clade definitions.

## Testing and validation

One of our goals was to make Phyx files automatically testable for whether the implied logical expression for the phyloreference would, when applied to the original reference phylogeny, indeed resolve to the same clade as the one intended by the original author. To facilitate visually inspecting the resolution result, the Phyx format needs to be able to store phylogenies side-by-side with phyloreferences in formats commonly used by tree-rendering tools, in particular the Newick format. In addition, the format needs to store an annotation to the phylogeny to indicate which clade the original author intended to define, and several related metadata, including a citation to the source of the phylogeny, and any curator notes about the phylogeny and the expected resolution, such as documenting clarifications about taxon substitutions made if one or more of the exact specifiers used in the clade definition are not present in the original phylogeny as a leaf node annotation.

For implementing an automated test suite, our aim was that it could perform tests at three main levels of abstraction:

1. At the level of unit tests, testing whether individual components of the Phyx format, such as taxonomic units, citations and phylogenies are represented syntactically correct.
2. At the level of integration tests, testing whether entire Phyx files (*i.e.,* digitized phyloreference records) could be interpreted and processed correctly as an OWL ontology and through OWL reasoning.
3. At the level of data validation, testing that a schema describing the Phyx format in the JSON Schema language can be used to correctly identify well-formed Phyx files and to provide useful errors when presented with badly-formed Phyx files.

Finally, we aimed to test that resolution of a particular phyloreference does not depend in some way on the presence or absence of certain tree topologies, of which the original reference phylogeny presents only one possibility. To ensure that resolution can identify clades across every possible tree topology, we generate every possible tree topology where the number of unique tips was between $n = 2$ (1 possible binary rooted tree, 0 possible multifurcating trees) to $n = 6$ (945 possible binary rooted trees, 1,807 possible multifurcating trees (see *Balding, Bishop & Cannings, 2007*)), and test whether simple minimum and maximum clade definitions resolve correctly on all topologies.

## RESULTS

### The Phyx format

Every Phyx document is a JSON-LD document that uses the '@context' of a particular version of the Phyx specification (currently http://www.phyloref.org/phyx.js/context/v1.0.0/phyx.json). The Phyx document includes metadata that describe the document itself, as described in Table 1. The relationships and cross-references between different data elements in a Phyx document are shown in Fig. 2.

Phyx documents contain entities of four types, which we will define in the following sections:

- **Citations** are used to record references to literature or online resources. Both phylogenies and phyloreferences may include citations to their sources, and taxonomic concepts use citations to reference a particular circumscription.
- **Taxonomic units** are parts of taxa being referenced by phylogenies and phyloreferences.
- **Phylogenies** are evolutionary hypotheses used for testing phyloreferences.
- **Phyloreferences** are computable clade definitions. They are primarily defined in terms of **specifiers**, which are taxonomic units that either must be included or must be excluded from the clade being defined.

### Citations

Citations are widely used to ensure that the sources of phylogenies and clade definitions are recorded. We created a standard type for citations based on the BibJSON format (http://okfnlabs.org/bibjson/), which is itself based on the BibTeX format (*Patashnik, 1988*). While none of these fields are required, Phyx editors should ensure that URLs and DOIs are included so that relevant documents can be accessed quickly and without ambiguity. Citations are described in Table S1.

**Table 1  Fields in the Phyx document.** Fields indicated with * are for use by supporting software (such as phyx.js) but are not mapped to RDF properties, and will thus not be converted into RDF.

| Field name | Description | Type | Example |
|---|---|---|---|
| @context | **Required**. The JSON-LD context, necessary to interpret this JSON file as an RDF file. | IRI | Depends on the version of the Phyx standard being used. Currently, this should be http://www.phyloref.org/phyx.js/context/v1.0.0/phyx.json |
| @type | Additional RDF types for the top level object. (“owl:Ontology” will be added automatically.) | Array of either IRIs or CURIEs | [ “ https://schema.org/DigitalDocument” ] |
| owl:imports | A list of OWL ontologies to be imported during reasoning. (See *Ontology metadata* for the list of ontologies that will be automatically added to this field.) | Array of IRIs | [ “ http://purl.obolibrary.org/obo/pato.owl” ] |
| doi | The Digital Object Identifier (doi) for this Phyx file. | DOI | 10.5281/zenodo.4562685 |
| source | A citation to this Phyx file. | Citation | See example of Citation below. |
| defaultNomenclaturalCodeIRI* | The default nomenclatural code to be used in this file, for both phylogenies and phyloreferences. This will only be used for nodes and taxon concept specifiers that don’t have a nomenclatural code set. | IRI | http://rs.tdwg.org/ontology/voc/TaxonName#ICZN |
| phylogenies | A list of phylogenies in this Phyx file. | Array of Phylogenies | See example of Phylogeny below. |
| phylorefs | A list of phyloreferences in this Phyx file. | Array of Phyloreferences | See example of Phyloreference below. |

## Taxonomic units

A taxonomic unit describes a unit of taxonomy that can be matched somewhere on a phylogeny. Three types of taxonomic units can currently be described using the Phyx format.

## IRI

Any taxonomic unit may be described with a single field called ‘@id’ with an IRI that points to a definition of a taxonomic unit. This allows complex taxonomic units to be defined in any way the user wants.

## Taxon or taxon concept

A taxon concept is a taxonomic grouping defined and described by taxonomists (*Kennedy et al. (2006)*, identified with a scientific name described under a particular nomenclatural code (for example, the International Code of Zoological Nomenclature). Taxon concepts not identifiable in this way may reference a specific circumscription by means of a citation to a publication in which that circumscription is defined (a “potential taxa” *sensu Berendsohn, 1995*), conventionally specified after the taxon name as a *secundus* (abbreviated to “sec.”). This can be specified in Phyx by using the *nameAccordingTo* field. Where no explicit circumscription is provided, the “nominal taxon concept” (*sensu Kennedy et al. (2006)*) will be assumed. The Phyx format supports both explicit and implicit circumscriptions.

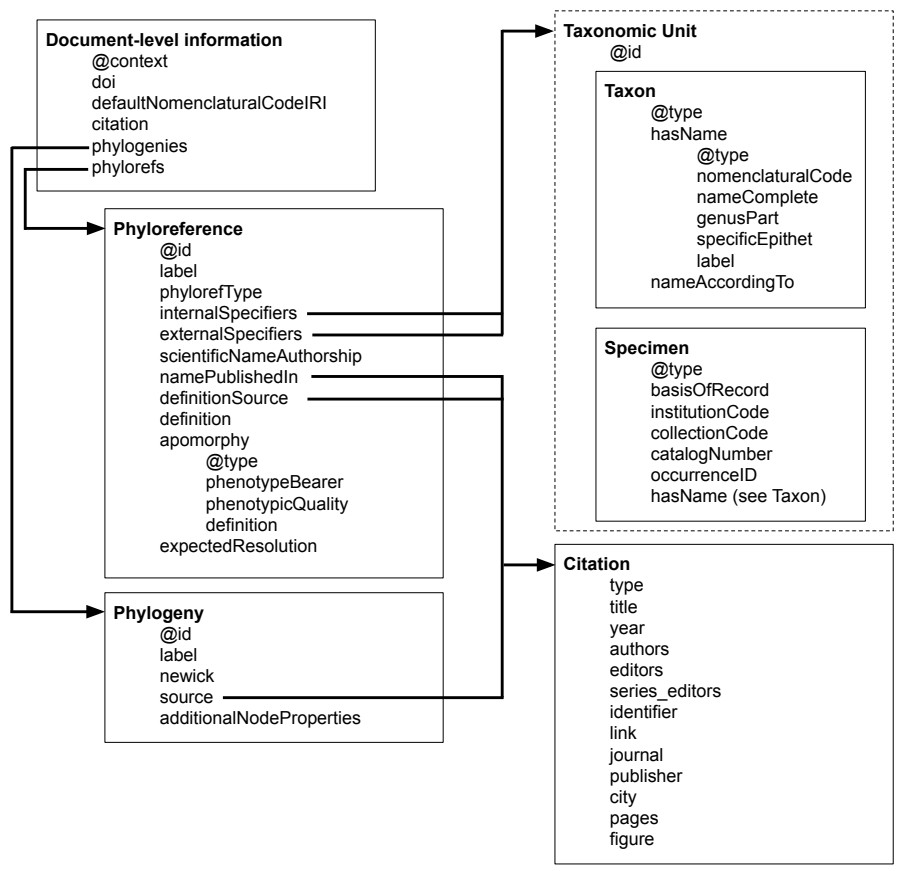

**Figure 2 Relationships and references between different types in the Phyx format.** Note that not all citation objects are fully expanded.

## Specimen

A *specimen* can be described using specimen identifiers following the Darwin Core (*Wieczorek et al., 2012*) term *occurrenceID* (http://rs.tdwg.org/dwc/terms/occurrenceID). If not a globally unique identifier (GUID), this is often constructed from the so-called Darwin Core Triplet, which is composed of the institution (*e.g.,* museum) code, the code of the collection within the institution, and the code of the specimen within the collection ("catalog number") (*Biodiversity Information Standards, TDWG)(2016*, sec. 2.7.3). Specimens are described in Table S2.

## Phylogenies

Phyx files may contain rooted phylogenies, *i.e.,* a tree-based representation of the evolutionary relationships between units of taxonomy. Each phylogeny must include the tree topology as a Newick string, stored in the 'newick' property. The Newick format allows both internal and terminal nodes to be labeled. It can also express polytomies, which are fully supported by the accompanying phyx.js programming library. Extended Newick formats for expressing phylogenetic networks (for example *Than, Ruths & Nakhleh, 2008*) resulting from, *e.g.,* reticulation events are, however, not currently supported by phyx.js.

**Table 2** Fields in taxonomic unit objects identified solely by IRI.

| Field name | Description | Type | Example |
|---|---|---|---|
| @id | **Required.** An opaque identifier for this taxonomic unit. | IRI | https://avibase.bsc-eoc.org/species.jsp?avibaseid=C600B65C997827CD (referring to a particular circumscription of a taxon concept labeled the "Canada or Cackling Goose") |

Each phylogeny can also include a label and a full citation to the published version using the 'citation' property.

When converted to an OWL ontology, each node in the phylogeny will be represented by a named individual of the class cdao:Node in the Comparative Data Analysis Ontology (CDAO) (*Prosdocimi et al., 2009*) data model. To add additional properties to nodes in the phylogeny, the *additionalProperties* field may be used. This is a dictionary mapping node labels to an object whose properties are copied into the Node object once it has been created. This can be used for example to indicate that a single node in the phylogeny refers to multiple taxonomic units by using *cdao:represents_TU* (see example in the table below).

## Phyloreferences

Phyloreferences are digitized clade definitions that capture in a structured and machine-processable way the semantics of the clade concept. To generate phyloreferences as OWL classes with formal logic definitions from the Phyx format, depending on the type of phyloreference two to three pieces of information are necessary:

1. The type of the clade definition (see Fig. 1).
2. One or more internal specifiers: the taxonomic unit(s) that must be included within the defined clade. Apomorphy-based phyloreferences must give exactly one.
3. Maximum-clade phyloreferences require one or more external specifiers: the taxonomic unit(s) that must be excluded from the defined clade.
4. Apomorphy-based phyloreferences must have exactly one apomorphy, exhibited by members of the taxonomic unit represented by the internal specifier.

Phyloreferences can also include additional metadata, such as a label for the phyloreference and citations for the clade definition as well as for the name (if the name is published according to a nomenclatural code). To store information on where the original author expected the clade definition to resolve on the reference phylogeny, each phyloreference can include an 'expectedResolution' dictionary. The keys of this dictionary are IRIs that refer to phylogenies, which must be present in the same Phyx document.

## Converting Phyx files to OWL ontology

Since Phyx documents are JSON-LD documents, they can be converted to other RDF formats by JSON-LD tools such as *rdfpipe* (https://rdflib.readthedocs.io/). In the following sections, we use a number of RDF prefixes to refer to terms in various ontologies. We list these prefixes in Table 6. The mapping of terms to RDF properties is defined by the Phyx JSON-LD context as given above; it is also reproduced in expanded form in Table S3 alongside the columns from Tables 1 to 5 and Tables S1–S2.

To be readable as OWL ontologies, some additional steps are necessary:

**Table 3 Fields in a taxon or taxon concept object.**

| Field name | Description | Type | Example |
|---|---|---|---|
| @type | **Required.** We use the TDWG Taxon Concept ontology (http://rs.tdwg.org/ontology/voc/TaxonConcept) for types and properties. | IRI | Must be http://rs.tdwg.org/ontology/voc/TaxonConcept#TaxonConcept |
| hasName | **Required.** The taxonomic name of this taxon or taxon concept. It is described with the following fields: | Object | |
| - @type | **Required.** We use the TDWG Taxon Name ontology (http://rs.tdwg.org/ontology/voc/TaxonName) for types and properties. | IRI | Must be http://rs.tdwg.org/ontology/voc/TaxonName#TaxonName |
| - nomenclaturalCode | The nomenclatural code under which this taxon name was created. | IRI | http://rs.tdwg.org/ontology/voc/TaxonName#ICZN |
| - nameComplete | **Required.** The complete uninomial, binomial or trinomial name without any authority or year components. | String | Alligator mississippiensis |
| - genusPart | The genus portion of the taxon name. | String | Alligator |
| - specificEpithet | The specific epithet portion of the taxon name. | String | mississippiensis |
| - label | The full taxon name, including an authority or year components. | String | Alligator mississippiensis (Daudin, 1802) |
| nameAccordingTo | Publication or authors whose circumscription of the taxon is intended to be used. If omitted, the nominal taxon concept (in the sense of *Kennedy et al., 2006*) will be assumed. | String | *Brochu (2003)* |

- Adding metadata on the ontology itself, such as which other ontologies to include.
- Converting all taxonomic units (on both phyloreferences and phylogenies) to OWL logical expressions.
- Converting phylogenies into an OWL representation.
- Converting phyloreferences into OWL logical expressions.

Below we detail each of these steps.

## Ontology metadata

A Phyx document will not be recognized as an OWL ontology, because it is not typed as 'owl:Ontology'. Therefore, conversion to OWL must include the following:

- Setting the document's '@type' JSON-LD property to an OWL ontology ('owl:Ontology').
- If reasoning needs to be enabled, then two application ontologies from the Phyloreferencing project must be included (using 'owl:imports'): 'http://ontology.phyloref.org/phyloref.owl' and (if taxonomic units are used) 'http://ontology.phyloref.org/tcan.owl'. These ontologies include the other ontologies mentioned in this file, such as CDAO, the TDWG ontology, NOMEN and others.

**Table 4  Fields in a phylogeny object.** Fields marked with * are for use by supporting software (such as phyx.js) but are not mapped to RDF properties, and will thus not be converted into RDF (or OWL).

| Field name | Description | Type | Example |
|---|---|---|---|
| @id | The identifier for this phylogeny. | IRI | #phylogeny0 |
| label | A label describing this phylogeny. | String | Fig 1 from *Brochu (2003)* |
| newick | **Required.** The Newick string. | String | (Parasuchia,(rauisuchians,Aetosauria,(sphenosuchians,(protosuchians,(mesosuchians,(Hylaeochampsa, Aegyptosuchus,Stomatosuchus,(Allodaposuchus,('Gavialis gangeticus',((('Diplocynodon ratelii',('Alligator mississippiensis','Caiman crocodilus')Alligatoridae)Alligatoroidea,('Tomistoma schlegelii',('Osteolaemus tetraspis','Crocodylus niloticus')Crocodylinae)Crocodylidae)Brevirostres) Crocodylia))Eusuchia)Mesoeucrocodylia) Crocodyliformes)Crocodylomorpha)) |
| source | The source of this phylogeny. | Citation | See above for an example Citation. |
| additionalNodeProperties* | A dictionary mapping node labels to properties that should be added to the node when converting this Phyx file into an OWL ontology. | Object | "additionalNodeProperties": { "Exodictyon incrassatum": { "representsTaxonomicUnits": [{ "@type": "http://rs.tdwg.org/dwc/terms/Occurrence", "institutionCode": "UC", "catalogNumber": "Wall 2527, Fiji" }] } } |

## Convert specifiers to OWL logical expressions

All taxonomic units can be converted into OWL logical expressions as below:

1. Taxon concepts are converted into an OWL expression in the form 'tc:hasName **some** (tn:nomenclaturalCode **value** tn:ICZN **and** tc:nameComplete **value** 'scientific name'). The 'tn:nomenclaturalCode' property can be removed if no nomenclatural code is specified. If the optional citation is a publication in which this taxon concept is circumscribed, the additional specification '**and** tc:accordingTo 'circumscription citation' may be added to this expression.

2. Specimen identifiers are converted into an OWL expression in the form 'dwc:occurrenceId **value** 'occurrenceID''. See the "occurrenceID" field in the Specimen description above for how this is constructed from specimen record properties if a native identifier is not present.

3. Apomorphies are not currently supported for converting to an OWL expression accessible to reasoning.

4. Other phyloreferences can be referenced by using their '@id'.

## Phylogenies

In order to connect phyloreferences to phylogenies, we reuse the concept of "taxonomic units" from the CDAO ontology (*Prosdocimi et al., 2009*). We use *CDAO:0000187*

**Table 5 Fields in a phyloreference object.** Fields indicated with * are for use by supporting software (such as phyx.js) but are not mapped to RDF properties, and will thus not be converted into RDF (or OWL). The phyx.js software copies the value of the 'phylorefType' field into the 'rdf:type' property during transformation to RDF.

| Field name | Description | Type | Example |
|---|---|---|---|
| @id | The identifier for this phyloreference. | IRI | #Alligatoroidea |
| label | A name for the clade defined by the phyloreference. For clade definitions digitized from the PhyloCode, the name will follow PhyloCode naming conventions (*De Queiroz & Cantino, 2020*). | String | Alligatoroidea |
| phylorefType | The type of this phyloreference. | Enumeration | One of phyloref:PhyloreferenceUsing Maximum Clade, phyloref:PhyloreferenceUsing MinimumClade or phyloref:PhyloreferenceUsing Apomorphy |
| scientificNameAuthorship | The authors who created this clade definition. | Citation | See the Citations section above for an example citation. |
| namePublishedIn | If the label is a scientific name, then this field records the publication in which that name was first published. | Citation | See the Citations section above for an example citation. |
| definitionSource | The publication in which this clade definition was first published. | Citation | See example citation above. |
| definition | **Required.** A free-text field for storing the verbatim clade definition. | String | Alligator mississippiensis and all crocodylians closer to it than to Crocodylus niloticus or Gavialis gangeticus. |
| internalSpecifiers* | A list of internal specifiers (defined as taxonomic units) that must be included in the clade. | Array of taxonomic units | See taxonomic unit examples above. |
| externalSpecifiers* | A list of external specifiers (defined as taxonomic units) that must be excluded from the clade. | Array of taxonomic units | See taxonomic unit examples above. |
| apomorphy | If used, indicates that this phyloreference designates the clade that arises from the first appearance of this trait that is synapomorphic with an internal specifier. In this case, exactly one internal specifier and no external specifiers must be provided. The trait is described with the following fields: | Object | |
| - @type | **Required.** Used to indicate the type of this trait. | IRI | Must be https://semanticscience.org/resource/SIO_010056 ("phenotype") |
| - bearingEntity* | An IRI that identifies the entity bearing the phenotypic quality if the phenotype referenced by this apomorphy can be decomposed into a quality and the entity bearing the quality (EQ model). | IRI | http://purl.obolibrary.org/obo/UBERON_0008271 |

**Table 5** (*continued*)

| Field name | Description | Type | Example |
|---|---|---|---|
| - phenotypicQuality* | An opaque IRI that identifies the phenotypic quality if the phenotype referenced by this apomorphy can be decomposed into entity and quality (EQ model). See *bearingEntity*. | IRI | Defaults to http://purl.obolibrary.org/obo/PATO_0000467 if one is not provided. |
| - definition | **Required.** A definition of the apomorphy. | String | A complete turtle shell as inherited by *Testudo graeca* |
| expectedResolution* | A dictionary of phylogeny identifiers to objects that record the nodeLabel (the node label on that phylogeny this phylogeny is expected to resolve to) as well as an optional description (describing why that node was chosen). | Dictionary | "expectedResolution": { "#phylogeny0": { "nodeLabel": "Gavialis gangeticus", "description": "Only representative of Gavialoidea in this phylogeny." } } |

**Table 6   RDF prefixes used in phyloreferencing.**

| Prefix | Ontology or vocabulary | Expands to |
|---|---|---|
| rdf | Resource Description Framework (RDF) | http://www.w3.org/1999/02/22-rdf-syntax-ns# |
| owl | Web Ontology Language (OWL) | http://www.w3.org/2002/07/owl# |
| CDAO | Comparative Data Analysis Ontology (CDAO) (*Prosdocimi et al., 2009*) | http://purl.obolibrary.org/obo/CDAO_ |
| dwc | Darwin Core (*Wieczorek et al., 2012*) | http://rs.tdwg.org/dwc/terms/ |
| tc | TDWG Taxon Concept LSID Ontology | http://rs.tdwg.org/ontology/voc/TaxonConcept# |
| tn | TDWG Taxon Name LSID Ontology | http://rs.tdwg.org/ontology/voc/TaxonName# |
| phyloref | The Phyloreferencing Ontology | http://ontology.phyloref.org/phyloref.owl# |
| tcan | Ontology for Taxon Concepts And Names | http://ontology.phyloref.org/tcan.owl# |

("represents_TU") to indicate that a particular node in the phylogeny represents a particular taxonomic unit. For example, we might say that ':node1 rdf:type [CDAO:0000187 ("represents_TU") **some** (tc:hasName **some** (tn:nomenclaturalCode **value** tn:ICZN **and** tc:nameComplete **value** '<scientific name>'))]".

## Phyloreferences

We use the following algorithm to transform the information in a Phyx file to generate OWL class expressions for logically defining the phyloreference class. This algorithm is implemented in and thus published in the phyx.js library (see below).

To generate the OWL model of a phyloreference, we utilize a series of properties and other terms drawn from the Phyloreferencing Ontology (prefix "phyloref:", see Table 6). Fully explaining and discussing the semantics of these terms is the subject of a future paper and beyond the scope of this one.

1. *phyloref:has_Sibling* is a symmetrical property that relates a phylogeny node to all other children of its parent, *i.e.,* if 'node1' and 'node2' share a parent, then 'node1 *phyloref:has_Sibling* node2' and 'node2 *phyloref:has_Sibling* node1'.

2. *phyloref:excludes_lineage_to* is a property defined such that if 'node1 *phyloref:excludes_lineage_to* node2' then node2 is a sibling of node1, or node2 is a

descendant (given by property CDAO:0000174, "has_Descendant") of a sibling of node1.

3. *phyloref:excludes_TU* is defined such that 'node1 *phyloref:excludes_TU* otu1' if node1 has a sibling that represents TU (property CDAO:0000187, "represents_TU") otu1, or if a descendant of a sibling of node1 does. In practice, this is a convenience property, defined as a property chain over phyloref:excludes_lineage_to, to make OWL restrictions underlying phyloreferences more readable.

4. *phyloref:includes_TU* is defined such that 'node1 *phyloref:includes_TU* otu1' if node1 represents TU (property CDAO:0000187, "represents_TU") otu1, or if a descendant of node1 (given by property CDAO:0000174, "has_Descendant") does.

**Phyloreference with one internal and one external specifiers:** This is the simplest OWL property restriction to generate for a phyloreference. It is true (which we call "resolves to") for the node that includes a particular taxonomic unit (internal1) and excludes another taxonomic unit (external1). Using the terms defined above, this can be expressed as: 'phyloref:includes_TU **some** internal1 **and** phyloref:excludes_TU **some** external2'.

**Phyloreference with two internal specifiers:** As per the semantics of phylogenetic definitions, this is equivalent to the parent of the node that includes one of the specifiers and through its sibling excludes the other specifier. Using 'internal1' and 'internal2' as specifiers, this can therefore be defined as 'CDAO:0000149 ("has_Child") **some** (phyloref:includes_TU **some** internal1 **and** phyloref:excludes_TU **some** internal2)". Because the phyloref:has_Sibling property is symmetric, it does not matter which of the two specifiers is chosen for inclusion, and which for exclusion.

**Phyloreference with more than two specifiers:** Phyloreferences with more than two specifiers can be converted into OWL property restrictions using the algorithm described below. To avoid repetition of complex logical expressions and to improve debugging by allowing resolution of components of the phyloreference to be examined, we create in many cases OWL classes defined as equivalent to components (*i.e,* parts) of the full logical expressions of a phyloreference. We refer to these below as component classes. For example, when defining a phyloreference with one internal specifier (*A*) and two external specifiers (*B* and *C*), we would create a component class each for logical expressions that resolve to (includes: *A*, excludes: *B*) and (includes: *A*, excludes: *C*). While describing the algorithm we use to generate the phyloreferences below, we indicate where these component classes are created.

*Minimum clade phyloreferences with multiple internal specifiers*: Generate logical expressions using the following recursive algorithm:

1. Define *createClassExpressionsForInternals* as a function that takes two arguments:
   a. *selected*: The set of chosen internal specifiers,
   b. *remainingInternals*: The set of internal specifiers that remain to be chosen.
   This function consists of the following steps:
   a. Generate a class expression for the provided sets of selected and remaining internal specifiers as follows:
      i. Generate *selectedExpr*, the logical expression for the internals that have already been selected:

1. If only a single internal specifier has been selected (*specifier1*) in *selected*, generate 'phyloref:includes_TU **some** *specifier1*''.
2. If two internal specifiers have been selected (*specifier1*, *specifier2*) in *selected*, generate the expression for a phyloreference with two internal specifiers: 'CDAO:0000149 ("has_Child") **some** (phyloref:includes_TU **some** *specifier2* **and** phyloref:excludes_TU **some** *specifier1*').
3. If *selected* consists of more than two internal specifiers, generate a component class for this logical expression by calling *createClassExpressionsForInternals(selected*, []).

  ii. Generate *remainingInternalsExpr*, the logical expression for the internals that remain to be selected:

    1. If only a single internal specifier remains to be selected (*specifier1*) in *remainingInternals*, generate 'phyloref:includes_TU **some** *specifier1* '.
    2. If two internal specifiers remain to be selected (*specifier1*, *specifier2*) in *remainingInternals*, generate the expression for a phyloreference with two internal specifiers: 'CDAO:0000149 ("has_Child") **some** (phyloref:includes_TU **some** *specifier2* **and** phyloref:excludes_TU **some** *specifier1*)'.
    3. If *remainingInternals* consists of more than two internal specifiers, generate a component class for this logical expression by recursively calling *createClassExpressionsForInternals(remainingInternals*, []*)*.

  iii. Generate a class expression in the form 'CDAO:0000149 ("has_Child") **some** (selectedExpr **and** phyloref:excludes_lineage_to **some** remainingInternalsExpr)'.

b. Generate class expressions for every combination of remaining specifiers as follows:

  i. Stop if there is only a single *remainingInternal* (the class expressions generated above fully cover this case), or if the size of the set *remainingInternal* is greater than the size of the set *selected* (otherwise previously generated class expressions would be duplicated).

  ii. For every remaining internal specifier *remaining1*, we generate a list of class expressions by calling *createClassExpressionsForInternals(selected* union *remaining1*, *remainingInternals* minus *remaining1)*.

c. Return the union of the class expressions generated in steps (a) and (b).

2. To generate the set of logical expressions for a phyloreference with a set of internal specifiers *internalSpecifiers*, we invoke the createClassExpressionsForInternals() function with the empty set as the selected specifiers: *createClassExpressionsForInternals(* [], *internalSpecifiers)*.

3. The phyloreference is equivalent to the union of the generated logical expressions, of which there will be $n$ $(n-1)/2$ with $n$ being the number of internal specifiers. However, logical disjunction is outside of the OWL2 EL profile, and is hence not used by the highly efficient reasoners available for this profile. To stay within the OWL2 EL profile, we create a separate subclass of this phyloreference as a component class for each

logical expression. Due to OWL subclass semantics, any nodes resolved by any of these subclasses are necessarily a valid resolution of their phyloreference superclass.

*Maximum clade phyloreferences using multiple internal and/or external specifiers*: We generate a logical expression for such phyloreferences using the following algorithm:

1. For every external specifier *selectedExternal*, generate an OWL restriction that is the intersection of:
   a. All the internal specifiers (in the form 'phyloref:includes_TU **some** *specifier* ').
   b. The selected external specifier (in the form 'phyloref:excludes_TU **some** *selectedExternal* ').
   c. All the external specifiers except for *selectedExternal,* in the form 'CDAO:0000144 ("has_Ancestor") **some** (phyloref:excludes_TU **some** *externalSpecifier*)'.
2. The phyloreference is equivalent to the union of the generated logical expressions, of which there will be as many as there are external specifiers. As noted for minimum clade phylorferences, to remain within OWL2 EL we instead create a separate subclass of this phyloreference as a component class for each logical expression.

## phyx.js library

The implementation of the conversion algorithm described above is included in a supporting JavaScript library we created, called phyx.js. It is available as open source code under the MIT license on Github at http://github.com/phyloref/phyx.js, and is published on NPM, the JavaScript package repository, at https://www.npmjs.com/package/@phyloref/phyx. In addition to the conversion algorithm, this library includes wrapper classes for most of the Phyx file components described above. The wrapper classes are not intended to completely abstract these elements, but to provide helper functions to help access them in standardized ways. Most wrapper classes are low-level, wrapping a single citation, taxon name or specimen. Intermediate wrapper classes wrap entire phylogenies and phyloreferences. A single high-level class wraps an entire Phyx document, and provides methods to convert it into an OWL ontology in JSON-LD when needed.

There are three sets of classes included in this library:

- The 'utils' classes are helper classes for the other parts of the library, for example a collection of OWL term names needed across the library ('owlterms.js'), and a simple cache to enhance the performance of classes that have to manage mappings ('PhyxCacheManager.js').
- The 'wrapper' classes wrap individual parts of the Phyx document. These are:
  - 'TaxonomicUnitWrapper' wraps any taxonomic unit. It can determine what type of taxonomic unit it has wrapped, and returns either a SpecimenWrapper or TaxonConceptWrapper depending on that type.
  - 'SpecimenWrapper' wraps taxonomic units that consist of a single specimen. It can return specimen identifier information, as well as the scientific name if this is also included.

- – 'TaxonConceptWrapper' wraps taxonomic units that consist of a taxon name and optionally a citation to a publication in which it is defined. It can wrap the taxon name using the TaxonNameWrapper class.
  - – 'TaxonNameWrapper' wraps a single taxon name. It provides methods to determine the name's nomenclatural code and to parse a string as a taxon name.
  - – 'PhylogenyWrapper': wraps a single phylogeny. It contains methods for converting the Newick string into its CDAO-based ontology form, as described above.
  - – 'PhylorefWrapper: wraps a single phyloreference. It contains methods for iterating over the internal and external specifiers, and for converting the phyloreferences into an ontological representation, as described above.
  - – 'PhyxWrapper' wraps an entire Phyx document and can convert it into an OWL ontology in JSON-LD as described above. It does this by adding the necessary metadata, and then invoking methods in the PhylorefWrapper and PhylogenyWrapper to wrap all phyloreferences and phylogenies.

- The 'matcher' classes provide methods for checking whether two different entities refer to the same entity. Currently, this includes

  - – The 'TaxonomicUnitMatcher' method, which provides three different methods for checking whether two taxonomic units refer to the same unit of taxonomy:
  - – matchByNameComplete() checks whether the two nameComplete fields in the taxonomic units are identical.
  - – matchByExternalReferences() checks whether the two '@id's for the two taxonomic units are identical.
  - – matchByOccurrenceID() checks whether the occurrence IDs –representing a specimen identifier –are identical.
  - – 'match()' attempts to use all three of the above methods to check whether the two taxonomic units match by any of the three criteria.

More formal descriptions of these methods and their arguments are available online at http://www.phyloref.org/phyx.js. The repository on GitHub includes an automatic test suite (*Taschuk & Wilson, 2017*), which provides low-level testing of components of this format to ensure that the library can parse a number of well-formed components and recognize a number of malformed ones. These include tests for phylogenies, phyloreferences, specimens, taxon names and taxonomic units. The repository also includes several example Phyx files, including one based on a set of clades published in *Brochu (2003)*, which is archived at Zenodo (https://doi.org/10.5281/zenodo.4562685), and a tutorial (see also Supplemental Information) that demonstrates major features using one of the example Phyx files. The test suite converts these example Phyx files into OWL ontologies expressed in the n-triples RDF format and confirms that they are identical to the expected output. We built a Java program for testing the resolution of these OWL ontologies, named JPhyloRef (*Vaidya, Cellinese & Lapp, 2021*). The test suite downloads JPhyloRef and tests the generated OWL ontologies to ensure that all phyloreferences contained in them resolve as expected, ensuring that the logical expressions were generated correctly.

## Systematic testing of phyloreference generation

In order to test whether the OWL expressions are being generated correctly, we generated every possible topology for phylogenies containing:

- 2 leaf nodes (1 topology)
- 3 leaf nodes (4 topologies, 3 of which are bifurcating)
- 4 leaf nodes (26 topologies, 15 of which are bifurcating)
- 5 leaf nodes (236 topologies, 105 of which are bifurcating)
- 6 leaf nodes (2,752 topologies, 945 of which are bifurcating)

We verified that the number of trees for a given number of leaf nodes matched the number expected (see *Balding, Bishop & Cannings, 2007*). To confirm that we had enumerated the topologies correctly, we wrote a Python script using the *DendroPy library* (*Sukumaran & Holder, 2010*) to check that every run yielded the expected number of bifurcating and multifurcating trees, that every tree had the expected number of leaf nodes, and that no two generated topologies were isomorphic as measured by a non-zero Robinson-Foulds distance (*Robinson & Foulds, 1981*).

For each topology, we named leaf nodes with letters from 'A' to 'F', and labeled every node in the tree based on which leaf nodes are descended from it, and, separated by an underscore, which nodes are descended from all of its sibling nodes, arranged in alphabetical order, starting with an initial letter "N". For example, given the tree (((A, B)*x*, C, D), E), the node *x* would be labeled as "NAB_CD" to indicate that the node's descendant leaf nodes are A and B, and that it has sibling nodes C and D. Note that the root node for every tree with five leaf nodes will be labeled "NABCDE_" under this scheme.

We then test phyloreference resolution on these topologies using two phyloreferences: a minimum clade phyloreference that resolves to the most recent common ancestor of 'A' and 'B', and a maximum clade phyloreference that resolves to the maximum clade that includes 'A' but excludes 'C'. Using the node labeling scheme described above, we can determine where each of these phyloreferences would be expected to resolve to:

1. The minimum clade example phyloreference should resolve to the node with the shortest label (*i.e.,* containing the fewest leaf nodes) starting with the characters "NAB".

2. The maximum clade example phyloreference should resolve to the node with the shortest label matching the regular expression pattern "NA.*_.*C.*" (where ".*" matches any character zero or more times).

We named the phyloreferences with the labels of the nodes where we expected them to resolve. We then converted the Phyx files into OWL ontologies and executed them using JPhyloRef (https://github.com/phyloref/jphyloref), a command-line Java wrapper for testing phyloreference resolution using the ELK reasoner (*Kazakov, Krötzsch & Simančík, 2014*). We found that for each topology ELK inferred the correct tree nodes as instances of the two phyloreferences.

The script for generating these phylogenies, named 'generate-topologies.js', is included in the Github repository, as are Nexus (*Maddison, Swofford & Maddison, 1997*) files containing the generated trees.

### phyx.js command line tools for end-users

The phyx.js package includes two command line tools suitable for end-users. One is 'phyx2owl.js', which converts Phyx files into OWL ontologies.

The other, 'resolve.js', resolves phyloreferences on the Open Tree of Life (*Hinchliff et al., 2015*). It is currently limited to phyloreferences whose specifiers are taxa identified with taxonomic names. The tool employs a two-step process: First, it uses the Open Tree of Life API to translate the taxonomic names of each specifier to their corresponding Open Tree Taxonomy Identifier (using the '/tnrs/match_names' endpoint). If any taxonomic name cannot be translated, the phyloreference is skipped. In the second step, it requests from the Open Tree of Life API to determine (*via* the '/tree_of_life/mrca' endpoint) the most recent common ancestor of all the internal specifiers, excluding, in the case of a maximum clade definition, all of the external specifiers. The response to this query is a list of nodes on the Open Tree of Life synthetic tree, starting with the earliest node that excludes the external nodes and the descendants between that node and the most recent common ancestor of the internal specifiers. For maximum clade phyloreferences it is therefore the first node in this list that is the correct ancestor node of the clade. For minimum clade phyloreferences, there will be only one node in the list, which will also be the correct ancestor node of the desired clade. 'resolve.js' will report the Open Tree of Life identifier for the ancestor node, as well as its node label if one is present.

'resolve.js' prints resolution results to standard output as a JSON document. The '–write-table' command line argument can also be used to write all resolution results to a tab-delimited file.

## DISCUSSION

In this paper we have described three main contributions: an exchange and archival format for phyloreferences called Phyx; a supporting software library written in JavaScript and published as open-source on the NPM package repository; and the testing framework we use to ensure that the content of Phyx files is syntactically correct and its semantics match expectations.

Among the key strengths of a phyloreference as the digitized form of a phylogenetic definition are that its semantics are computable, meaning that machines can digest and compute with the (evolutionary) meaning of the phylogenetic definition and apply it to phylogenetic hypotheses, *i.e.,* phylogenetic trees. Consequently, initially our digitization target was an OWL ontology. However, as explained earlier, it became clear that an OWL ontology as the direct product of digitizing a phylogenetic definition faces a number of problems, including a still relatively poor software support ecosystem; challenges with editing the serialization formats; and inherent limitations of OWL (and in fact any first-order logic system) for properly representing clade definitions utilizing more than the minimally required set of specifiers. As David Wheeler's famous aphorism, a.k.a. as the fundamental theorem of software engineering, goes "All problems in computer science can be solved by another level of indirection" (*Spinellis, 2007*). Indeed, our decision to devise Phyx as a distinct format allowed us to decouple the tasks of managing, exchanging,

and long-term archiving the products of digitizing phylogenetic definitions on the one hand, and applying machine reasoning to them on the other hand. As a result, using the Phyx format instead of an OWL ontology as the initial serialization product has greatly simplified the workflow of curating phyloreferences from published clade definitions, and the development of curator-supporting tools (which we will report in future papers). At the same time, as a JSON-LD format most content in a Phyx file remains tightly tied to ontology-defined semantics of properties and types, which has also allowed us to build a powerful testing framework that can catch syntactic as well as semantic errors early on.

Another consequence of devising the Phyx format as essentially a level of indirection is that by design we can record phyloreferences in the Phyx format that will be difficult to properly represent in an OWL ontology, as we explain in describing the algorithm that generates class equivalency expressions from the attributes of phyloreferences in Phyx format. We consider this an advantage rather than a downside: it allows us to record in a fully structured, semantically explicit, and machine parseable way the information about a phylogenetic definition without having to commit to it having to be representable in OWL or more generally in some first-order logic language. In this way, Phyx allows for lossless management and archival of the information, while allowing for different ways, including first-order logic reasoning, to be developed for resolving phyloreferences of various kinds and complexity.

The phylogenetic hypotheses to which one may want to apply a phylogenetic definition may contain polytomies (multifurcations), for example, as a result of insufficient resolution of the branching order, and they can also include reticulation events, representing processes such as hybridization, lateral gene transfer, *etc*. Phylogenetic definitions are agnostic to both polytomies and the presence of reticulation, and thus, by extension, so is Phyx as a machine-readable data standard. When reticulation events are present in a phylogeny, which is then often referred to as a phylogenetic network, some clades will overlap with others rather than be mutually exclusive. The PhyloCode as the nomenclator for phylogenetic definitions explicitly allows this, and it does not alter the formalism and elements from which a phylogenetic definition is constructed. Whether the presence of reticulation events in a phylogeny presents an issue for computational resolution of a phyloreference will necessarily depend on the chosen mechanism, which Phyx as the data standard intentionally does not prescribe. Answering this question is therefore beyond the scope of this paper, and is a subject for future work.

Phyx is not the first approach to digitizing phylogenetic definitions. Earlier efforts include in particular mor (*Hibbett et al., 2005*) and Names On Nodes (*Keesey, 2007*), and more recently the Phlora application (http://phlora.org/) allows clades to be visualized on a tree. Names on Nodes uses MathML (https://www.w3.org/Math) (an XML format), which in contrast to JSON-LD is not well supported by software tooling, arguably does not qualify as human-readable text, and lacks a mechanism for tight integration with formal semantics in the form of ontologies. The format itself also does not cover the various structured metadata we describe for Phyx, and instead focuses on how to define different groups of organisms (not all of which form clades). Other efforts (such as mor and Phlora)

provide only a limited means of expressing a clade definition, and don't publish a more encompassing format specification.

## CONCLUSIONS

We present Phyx, an interoperable exchange and archival format for digitized clade definitions (called phyloreferences). The format accommodates not only the phylogenetic definition itself, but also a comprehensive set of metadata and an optionally accompanying reference phylogeny. By virtue of being based on JSON-LD, the data and metadata it represents are tightly linked to well-defined semantics, allowing the semantics of the clade definition to be fully computable, such as in the form of an OWL ontology. We also present phyx.js, a supporting software library in JavaScript that greatly simplifies programming common tasks in composing and managing Phyx documents in end-user applications. Other features of the software library include generating OWL ontologies from Phyx documents; an end-user application for resolving phyloreferences in Phyx files against the synthetic Open Tree of Life; and a testing framework to validate both the syntactic and semantic correctness of the content of Phyx files. Although not the first approach to digitizing clade definitions, Phyx is to the best of our knowledge the first digitization format for clade definitions that is comprehensive, maintains tight links to formal semantics, and is suitable for technology-agnostic permanent archival. At the same time, the accompanying software makes the format easy to work with for developers, while also supporting complex tasks such as converting Phyx documents into computable and testable OWL ontologies.

## ACKNOWLEDGEMENTS

We thank Daniel Caetano and an anonymous reviewer for helping to improve this manuscript through their thoughtful and constructive comments.

### Funding
This work was supported by the US National Science Foundation through collaborative grants DBI-1458484 and DBI-1458604. The funders had no role in study design, data collection and analysis, decision to publish, or preparation of the manuscript.

### Grant Disclosures
The following grant information was disclosed by the authors:
The US National Science Foundation through collaborative grants: DBI-1458484, DBI-1458604.

### Competing Interests
Hilmar Lapp is an Academic Editor for PeerJ.

## Author Contributions

- Gaurav Vaidya conceived and designed the experiments, performed the experiments, analyzed the data, prepared figures and/or tables, authored or reviewed drafts of the paper, developed the Phyx format and wrote the supporting software, and approved the final draft.
- Nico Cellinese conceived and designed the experiments, authored or reviewed drafts of the paper, and approved the final draft.
- Hilmar Lapp conceived and designed the experiments, analyzed the data, authored or reviewed drafts of the paper, and approved the final draft.

## Data Availability

All software is available under the MIT License at https://github.com/phyloref/phyx.js. The release coinciding with this submission is v1.0.1 (https://github.com/phyloref/phyx.js/releases/tag/v1.0.1), and is archived at Zenodo (https://doi.org/10.5281/zenodo.5576557).

The *Brochu (2003)*-derived example Phyx file (and its conversion to RDF/OWL) is available on Zenodo at: Vaidya, Gaurav. (2021). Digital representation of some of the clade definitions in *Brochu (2003)* in the Phyloreference Exchange (Phyx) format (1.0.0) [Data set]. Zenodo. https://doi.org/10.5281/zenodo.4562685.

## Supplemental Information

Supplemental information for this article can be found online at http://dx.doi.org/10.7717/peerj.12618#supplemental-information.

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
