# Peer review of "A new phylogenetic data standard for computable clade definitions: the Phyloreference Exchange Format (Phyx)"

_PeerJ, doi:10.7717/peerj.12618_

## Round 0.1 · original submission · Minor Revisions

I have now received two reviews of your paper. Both reviewers found it informative and interesting, however, both also identified a number of areas for improvement. These all look very doable to me and I believe will help with the readability of the paper. Good luck with your revision.

Reviewer 1 ·

Basic reporting

no comment

Experimental design

no comment

Validity of the findings

no comment

Additional comments

The authors proposed a new data standard, the Phyx format, for computable clade definitions. IMO, the paper is too long and I would recommend the authors to move some of the tables and corresponding description to supplemental file and only highlight the most important and essential part of this format. This will make it more clear of the novelty and advantage of the format.

The authors developed phyx.js for the phyx format. However, I cannot find any tutorial for using it. It would be better to provide guidance in supplemental file as both the library and the format are new to everyone.

The author emphasize the advantage of this format is that the clade definition is computable. A case study to demonstrate the usage and possible application is useful especially for newly introduced format.

·

Basic reporting

The text is well-written. The subject is highly technical and dense, however, the authors wrote the text in a clear way. The manuscript provides plenty of references to solidify the claims made herein. All resources described in this manuscript are openly available to the community and can be used on any computer platform. The approach was described with sufficient detail and accuracy.

Experimental design

This manuscript does not aim at testing a particular hypothesis using experiments or meta-analyses. The manuscript provides a description of a bioinformatics resource. Evaluation of the experimental design does not apply.

Validity of the findings

The resource described by the authors has the potential to significantly improve the usability of the Tree of Life. The ability to reference clades given some pointers, such as two OTU names and a synapomorphy could change the way we manipulate the Tree of Life. More importantly, a resource like the one described here is the gateway to a fully tree-thinking paradigm in biology. Today, the first contact of students (and the general public) with biodiversity is associated with taxonomical rankings. Field guides, Wikipedia, TV shows, classic school books, all rely on taxonomic names to provide meaning and organization to biodiversity. Allowing for a simple and computer-readable organization of phylogenetic data can change that. In the future, we could be looking at a phylogenetic tree as the first result of an internet search for a taxon name. Students would learn to think using phylogeny and, as the authors mention, think about evolutionary groupings before thinking about character similarity. In summary, we have changed little with the introduction of bioinformatics. We still use static drawings (phylogenies in pdf files) and data tables (once in field notebooks and today in spreadsheet files---not very different). Storing and sharing phylogenies, taxon, and group definitions as well as data in a computer-readable way could open the avenue for more dynamic and interconnected research in evolutionary biology.

Additional comments

I read this manuscript from the perspective of a person who works with phylogenetic trees (either using trees from the literature or constructing new trees). At this moment, the most used workflow, as far as I know, is to store and share the phylogenies as newick or nexus files. When morphological, ecological, or behavioral data is available, this is stored as a data table (such as .csv) with the OTU names matching the ones in the newick or nexus files.

It seems to me that we (i.e., the community) could help by formatting/storing the phylogeny and associated data in a specific way. It would be helpful to add some directions aimed at the end-user (if that is the case). For example, when should we fill the phyx data fields? Should the researcher provide some file together with a publication or is this integrated into another website/software? Maybe a brief paragraph on the Introduction putting into context in which step the resource described here is to be used?

I could be confused and the resources described here are aimed specifically for software developers only. In that case, maybe this should be made more clear? I probably missed where this was explained.

Can this phylogenetic data standard deal with phylogenetic networks? What about soft and hard polytomies?

Specific comments:

Lines 8 to 9: Suggest to exclude “by far”. The word “most” already communicates how prevalent this is.
Lines 25 to 27: What about phylogenetic networks? Can an approach like this be useful when the phylogeny includes horizontal relationships due to hybridization, introgression, or horizontal gene transfer?
Line 30: “the wild” means “the literature”?
Lines 619 to 628: Traditionally, the Introduction is the place to present what has already been done in the field and previous papers/software/products that are related to the manuscript. I suggest moving the mention to these products to the Introduction.

---

## Round 0.2 · accepted · Accept

Thanks for the nice work! It's ready to be accepted.